# Assessing Feasibility of Water Resource Protection Practice at Catchment Level: A Case of the Blesbokspruit River Catchment, South Africa

Koleka Makanda [1,2,*], Stanley Nzama [3] and Thokozani Kanyerere [2]

1 Water Resource Classification, Department of Water and Sanitation, Pretoria 0001, South Africa
2 Department of Earth Sciences, University of the Western Cape, Cape Town 7535, South Africa; tkanyerere@uwc.ac.za
3 Reserve Determination, Department of Water and Sanitation, Pretoria 0001, South Africa; nzamas@dwa.gov.za
* Correspondence: cornie.makanda@gmail.com; Tel.: +27-123-368-406

**Abstract:** The operationalization of water resource protection initiatives for surface water resource quality and equitable water quality allocation is critical for sustainable socio-economic development. This paper assessed Blesbokspruit River Catchment's water quality status, using the South African Water Quality standards and Water Quality Index (WQI). Protection levels for quality, and waste discharge for point sources were set and evaluated using the total maximum daily loads (TMDLs) and chemical mass balance (CMB) techniques, respectively. The study found that the water quality results for the analysed physico-chemical parameters ($Na^+$, $Ca^{2+}$, $Mg^{2+}$, $Cl^-$, $F^-$, pH, EC, $SO_4^{2-}$) of the data collected from 2015 to 2022 were within the limits of the water quality standards, except for $NO_3^-$ and $PO_4^{2-}$. The water quality from the study area was categorized as acceptable for drinking purposes with the WQI of 54.80. The application of the TMDL approach resulted in the 77.96 mS/m for electrical conductivity (EC), 9.92 mg/L for phosphate ($PO_4^{2-}$), and 15.16 mg/L for nitrate $NO_3^-$ being set as the protection levels for the catchment. The CMB was found to be a useful tool for the evaluation of point source discharges into water resources. The study recommends the application of TMDL and CMB techniques in water resource protection practice.

**Keywords:** chemical mass balance; protection limits; total maximum daily loads; water; quality allocation; water quality index; water resource protection

## 1. Introduction

Water resources are crucial for sustaining a sufficient food supply, a fruitful habitat for biodiversity, and a healthy environment for all living things. Fundamentally, water is one of the most essential needs for life [1,2]. Globally, surface water quality has become a complex issue, one that remains very important for long-term economic development, welfare, and environmental viability [3,4]. Surface water quality, which is impacted by natural processes and anthropogenic activities [5–8], requires efficient protection for pollution prevention, especially in areas where freshwater is limited [9–13].

In South Africa, freshwater resources are the most essential resources for human existence and growth [14]. In the country, water quality challenges related to freshwater resources are well reported in the literature [15–17]. Such challenges linked to water quality deterioration were confirmed by [3], who reported that several water management areas in the country are experiencing water shortages and quality deterioration while natural systems are put under enormous pressure. Changes in river flow patterns have also been identified as one of the factors that negatively influence water quality, in addition to influences from anthropogenic activities [18]. Ref. [19] argued that the adoption of water resources management strategies that aim to strike a balance between water resource conservation and their sustainable utilization is essential considering that surface water

resources are susceptible to contamination and overexploitation due to their accessibility and fragility.

The South African water quality standards for drinking water [20] and other South African Water Quality Guidelines (SAWQG) for various water uses are available and extensively used in the county to ensure that their requirements in terms of water quality are met. Water resource assessment using water quality guidelines provides information on whether water from a source is suitable for meeting fundamental human needs, such as being fit for drinking, or any other water uses, and in cases of unsuitability treatments, processes can be recommended before consumption [21–23]. However, numerical limits prescribed in the water quality guidelines are the same nationally, and they are a requirement for a specific water use such as domestic applications, industrial applications, and in aquatic ecosystem. The numerical limits prescribed in the guidelines are user-specific, and do not necessarily reflect the spatial or temporal variability of a catchment. Therefore, in order to improve water resource protection at the catchment level, numerical limits for water quality formulated on the bases of the prevailing conditions of a particular catchment must be determined (catchment-specific protection levels). In the country, the protection of water resources is ensured by undertaking studies on resource- directed measures such as resource quality objectives (RQOs). The numerical limits for RQOs are set by taking into consideration the background conditions and spatial variability of individual catchments; hence, they differ from one catchment to another. Therefore, while water quality guidelines set numerical limits for various water uses, resource-directed measures (RDMs) such as RQOs set limits for a particular resource to protect its current conditions or improve them. In this regard, several studies on RDMs have been concluded in the country [24]. Results from such studies and projects involving resource-directed actions are reported in the official publications from the government. The published indicators and numerical limits for water quality are prescribed for implementation at the catchment level [25,26].

According to [17], one of the main pressing issues regarding water resource protection practices at the catchment level is a clear link between the water use license conditions set for users and numerical limits for water quality set as protection levels for a catchment. The understanding of how water resource protection limits for surface water quality and water use conditions are set in a catchment can significantly improve water resource protection practices at catchment level. Previously, researchers have used programming models (simulations) for the purpose of waste allocation taking into consideration established water resource protection requirements [27–33].

However, in the absence of more advanced modelling software to model scenarios of a system's water quality parameters in a catchment, the application of water resource protection initiatives at the basin level becomes a challenging task for water resource managers. At the Blesbokspruit River Catchment, there is limited knowledge of how to operationalize water resource protection initiatives set for the catchment, a situation which requires addressing. Therefore, the aim of this investigation was to showcase how water resource protection initiatives can be practically applied at a water resource level using the techniques of water quality index (WQI), total maximum daily loads (TMDLs), and chemical mass balance (CMB) through a case study of the Blesbokspruit River Catchment. For the desired outcome of the investigation to be achieved, the following specific objectives were established: (i) to assess and evaluate the surface water quality status of the Blesbokspruit River Catchment using WQI; (ii) to estimate the TMDL allowable for use as protection limits for surface water resources in the catchment; and (iii) to estimate the waste load allocation for surface water quality using the CMB model.

## 2. Materials and Methods

The methodological approach applied in the study is shown in Figure 1 and comprises the following: (1) data collection and analysis; (2) water quality index calculation; (3) total maximum daily load estimation; and (4) waste load allocation.

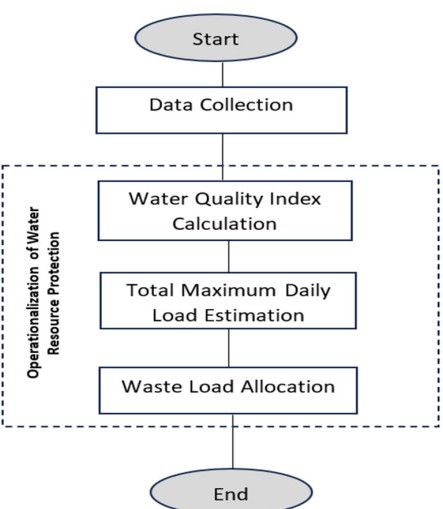

**Figure 1.** Flowchart for water resource protection practices in the Blesbokspruit River Catchment.

*2.1. Study Area Description*

The study area which is the Blesbokspruit River Catchment (BRC) is completely located to the south of the equator between latitudes 26°1.91′ S–26°36.97′ S and longitudes 28°23.41′ E–28°19.97′ E (Figure 2). The catchment area is within the Upper Vaal Water Management Area (UV-WMA) of the broader Vaal River System with the sub-catchments C21D, C21E, and C21F. The BRC, which is found 40 km to the south-east of Johannesburg, holds a delicate position within the Gauteng Province due toits location within the East Rand Region. The catchment is surrounded by 5 towns, namely: Springs, Benoni, Boksburg, Brakpan, and Nigel, which form part of South Africa's densely urbanized and industrialised hub [34]. The catchment drains into the Vaal River system which is the main source of water supply to the residents of Gauteng Province [3]. A considerable region of formal and informal urban development surrounds the basin. About 45% of the watershed is urbanized, with the remainder comprising mining, industrial, and agricultural activity. It has been reported that the quality of the BRC has dramatically deteriorated due to the discharge of mining effluent and sewerage, as well as other pollution linked with urbanization, industrialization, and agricultural growth in recent decades. [3,34]. The catchment was identified as Resource Unit 62 (RU62) within the delineated integrated unit of analysis UI of the UV-WMA [24]. The catchment was prioritised for water resource protection, and numerical limits for nitrate, phosphate, and electrical conductivity were prescribed and legalized through the government gazette [24]. This makes the BRC an ideal case study area for testing the feasibility of RDM operationalization.

*2.2. Data Collection*

This study relied on the secondary data sourced from the national water quality database called the Water Management System (WMS) of the South African Department of Water and Sanitation (DWS). This study utilised data from 2015 to 2022 from 9 in-stream assessment sites and examined 34 groundwater sites within the BRC; the sampling locations are shown in Figure 2a. These datasets were generated through the field data collection under the National Chemical Monitoring Programme (NCMP) of the DWS. The NCMP uses standard methods as outlined in [35,36] to collect data directly from the aquatic environment for surface water and from the field for groundwater. Briefly, water samples for physico-chemical analysis were collected from the sites within the main perennial Blesbokspruit tributary using the grab sampling technique [35]. Groundwater samples were obtained using a bailer from groundwater locations (boreholes) after boreholes had been purged and the water quality field characteristics (temperature, pH, electrical conductivity (EC)) had stabilized [21,36]. Water samples were subsequently put into the entire capacity of sterile 250 mL polyethylene sampling vials for reducing the headspace volume and were labelled accordingly. To protect

the samples from microbial activity that could result in changes in chemical constituents and concentrations within a water sample, one ampoule of mercury chloride (HgCl) was added to each sample bottle. Samples were packed in a cooler box with ice packs (to keep the temperatures low) and transported to the Resource Quality Information Services (RQIS) national laboratory for analysis. Samples were stored in a dark cooler room at a temperature below 4 °C until analysis was performed by the laboratory. The chemical analyses were undertaken using Aquakem 250 Photometric Analyzer, Flow Injection Analyzer (FIA), and Flammable Atomic Absorption Spectrophotometry (FAAS). Following that, the chemical analysis findings were recorded in the WMS database for future use.

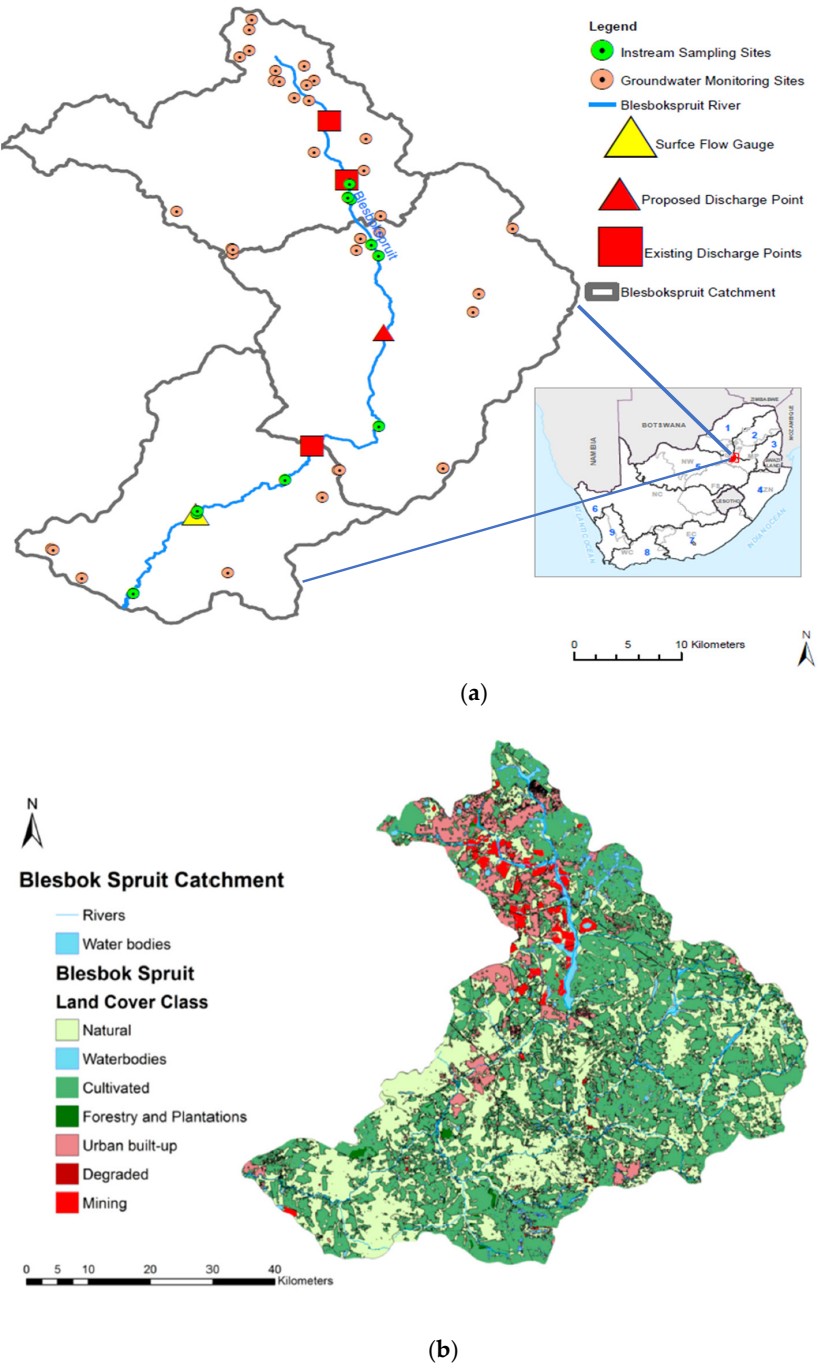

(a)

(b)

**Figure 2.** (**a**) Locality map of the Blesbokspruit River Catchment; and (**b**) Study area showing land cover activities, adopted from [3].

### 2.3. Data Analysis

Water quality data from the WMS database were transferred to the Microsoft Excel 2016 spreadsheet for descriptive statistical analysis. For each parameter, the annual averages and standard deviations were calculated. Table 1 displays the descriptive statistics (maximum, minimum, median, and standard deviation) for all 10 water quality parameters that were analysed. The water quality metrics evaluated for investigation include common main cations ($Na^+$, $Ca^{2+}$, $Mg^{2+}$), anions ($NO_3^-$, $Cl^-$, $F^-$, $SO_4^{2-}$ $PO_4^-$) and physical water quality parameters such as electrical conductivity (EC) and pH. The specified water quality parameters are critical for assessing drinking water quality in accordance with the national standards [19,20], and for aquatic ecosystem assessment [37].

**Table 1.** Water quality standards for domestic use and aquatic ecosystem [20,37].

| WQ Variables | Measurement Units | Class 0 | Class I | Class II | Class III |
|:---:|:---:|:---:|:---:|:---:|:---:|
| pH | pH units | 6–9 | 5–6 and 9–9.5 | 4–5 and >9.5–10 | <4 and >10 |
| EC | mS/m | <70 | 70–150 | 150–370 | >370 |
| Ca | mg/L | <80 | 80–150 | 150–300 | >300 |
| Cl | mg/L | <100 | 100–200 | 200–600 | >600 |
| F | mg/L | <0.7 | 0.7–1.0 | 1.0–1.5 | >1.5 |
| Mg | mg/L | <70 | 70–100 | 100–200 | >200 |
| $NO_3$ | mg/L | <6 | 6–10 | 10–20 | >20 |
| Na | mg/L | <100 | 100–200 | 200–400 | >400 |
| $SO_4$ | mg/L | <200 | 200–400 | 400–600 | >600 |
| $PO_4$ | mg/L | <0.005 | 0.005–0.025 | 0.025–0.25 | >0.25 |

### 2.4. Water Quality Index Calculation

In assessing the surface water quality status in the research area, the limits prescribed in the water quality standards (Table 1) were utilized as a measure of unprocessed water quality. Water quality variables are described as class 0, class I, class II, and class III. The water quality classes describe raw water that is considered to be ideal water for domestic use, acceptable water for domestic use, tolerable water for domestic use, and unacceptable water for domestic use, respectively. Therefore, the average values of the concentrations for the 10 water quality parameters were compared to the limits prescribed in the guidelines.

Analysed data were also used to establish monthly changes in the concentrations of water quality parameters during the period from 2015 to 2022. This was achieved by doing trend analysis, and monthly changes were presented in the graphs of monthly averages for each chemical or physical parameter. Trend analysis was performed to establish changes in the concentrations of water quality constituents during the 7-year period. In order to obtain a comprehensive picture of the overall surface water quality within the catchment, the water quality index (WQI) was used to establish the overall water quality status of the catchment. WQI is used to simplify and convey scientific water quality information by combining the influence of various water quality parameters into a single-digit score that describes the overall water quality in a watershed [38]. The index delivers meaningful and understandable water quality information to policymakers and the general public on river quality status with a scientific basis [21,39–41]. WQI models have four stages: (1) selecting the water quality parameters of interest; (2) generating sub-indices for each parameter; (3) calculating the parameter weighting values; and (4) aggregating sub-indices to compute the overall water quality index. All four stages are demonstrated in Figure 3. The WQI was calculated for the effective nine selected parameters of the surface water quality (pH, EC, Ca, Cl, F, Mg, $NO_3$, Na, $SO_4$) by following a five-step procedure as per [21,42].

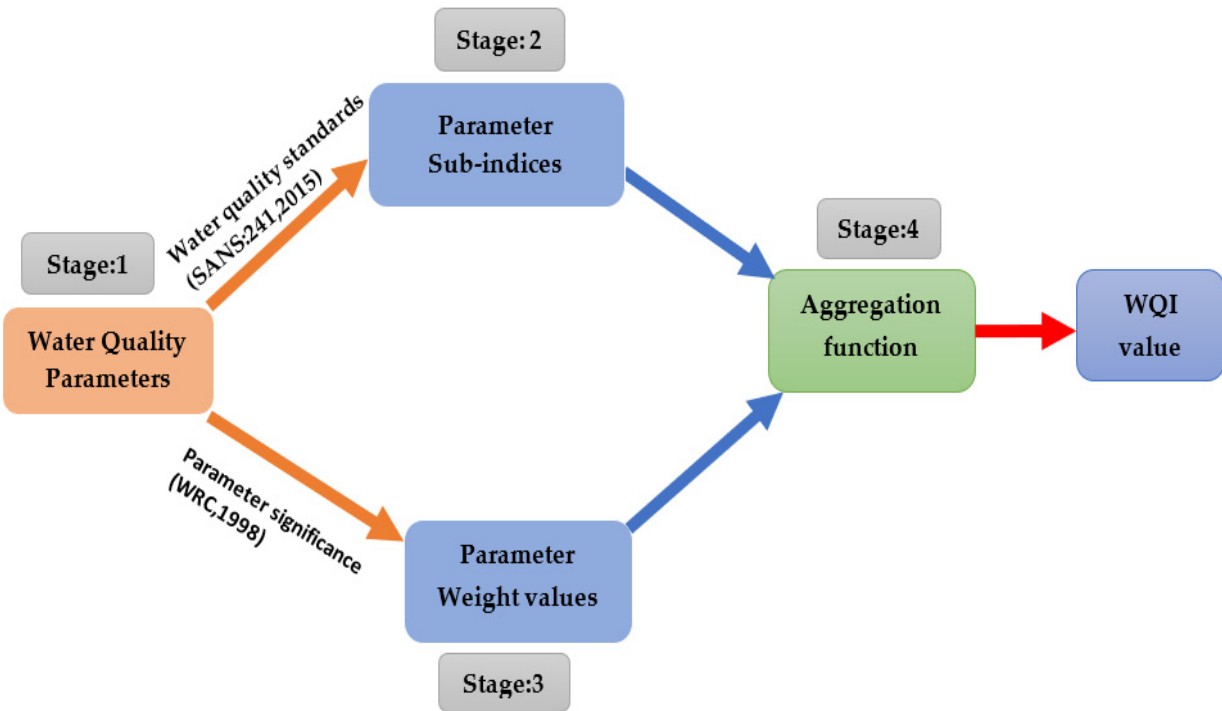

**Figure 3.** A general structure of the WQI model, adapted and modified from [38].

The initial step was to allocate weight (*wi*) to the chosen water quality parameters. According to [43,44] a weight value ranging from 1 to 5 was assigned to water quality parameters for estimating the water quality index, where nitrate is usually assigned with the maximum weight of 5. A weight of 4 was assigned to $Cl^-$, $F^-$, and $SO_4^{2-}$, with 3 assigned to the $Na^+$, $Ca^{2+}$ and $Mg^{2+}$, while pH and EC were assigned a weight of 2 according to [21]. In the second stage, Equation (1) was used to determine a relative weight (*Wi*) for each of the selected water quality parameters.

$$Wi = wi / \sum_{i=1}^{n} wi \tag{1}$$

The third stage was to calculate and assign a quality rating scale (*qi*) for each parameter by dividing the concentration of each water quality parameter (*Ci*) by the South African water quality standard for domestic use (*Si*). When the [20] guideline was applied, numerical limits for class I were considered, whereas when the [19] standard was utilized, numerical limits as specified in the standard were considered. Equation (2) was used to convert the results into percentages.

$$qi = \left(\frac{Ci}{Si}\right) \times 100 \tag{2}$$

The fourth stage involved calculating the sub-index (*Sl_i*) for each water quality metric using the algorithm in Equation (3).

$$SI_i = W_i \cdot q_i \tag{3}$$

The WQI for the entire study area was determined using Equation (4) in the fifth stage.

$$WQI = \sum SI_i \tag{4}$$

As shown in Table 2, the estimated WQI values for the research area were subsequently categorised into five types of water quality ratings [14,21,41] related to water quality classes [20].

**Table 2.** Water quality categorisation using water quality index [21] and South African water quality standards [19,20].

| WQI Ratings | Definition | Water Quality Class | Definition |
|:---:|:---:|:---:|:---:|
| <50 | Excellent | Class 0 | Ideal water for domestic use |
| 50–100 | Good | Class I | Acceptable water for domestic use |
| 100–200 | Poor | Class II | Tolerable water for domestic use |
| 200–300 | Very poor | Class III | Unacceptable water for domestic use |

*2.5. Total Maximum Daily Load Estimation for Water Resource Protection*

The protection limits for surface water quality in the study area were established by a simple total maximum daily loads (TMDL) approach based on the model by [45]. The model was created as a tool to assist in the restoration and protection of waterbodies where beneficial uses for aquatic life, recreation, public drinking water, or human health are impeded or threatened. Conceptually, the model is based on the fact that the pollutant loading of a water body originates from different pollutant sources such as point and non-point sources [46–48]. In the development process of TMDLs, the determination of a margin of safety (MOS) is necessary to account for any uncertainty regarding the relationship between pollution loads and receiving waterbody quality. [49]. In this current study, Equation (5) was used to establish surface water quality protection limits for EC, $NO_3$, and $PO_4$ in the study catchment.

$$TMDL = \Sigma WLA + \Sigma LA + MOS \tag{5}$$

where:

TMDL = Total maximum daily loads set as protection levels for the catchment (mg/s).
$\Sigma$WLA = Sum of waste load allocation to existing permitted point sources (mg/s).
$\Sigma$LA = Sum of load allocation for non-point sources (groundwater signature was used) (mg/s).
MOS = Margin of safety (set at 10% in this study) accounts for any uncertainty associated with attaining the protection levels for water quality.
Load = Discharge (L/s) × concentration (mg/L).
Load = mg/s.

The TMDL estimation was performed on the main river stem of the Blesbokspruit River Catchment. The estimation was based on the analysis of data derived from the instream sampling sites, discharge points, groundwater monitoring sites, and flow gauging station (B1H032) as shown in Figure 2. Data from the discharge points were used to estimate $\Sigma$WLA, while data from the instream sampling sites were used to obtain natural background levels. Data from the groundwater monitoring sites were used to estimate the levels of non-point sources. Altogether, data from the groundwater monitoring sites, instream sampling sites, and flow data were used to calculate $\Sigma$LA.

*2.6. Waste Load Allocation*

To demonstrate the effects of the newly set TMDLs on the evaluation of a new point discharge within the BRC, the chemical mass balance (CMB) approach was applied in a hypothetical case example. The existing point discharges survey was carried within the study area resulting in a presentation of a system diagram. This process was deemed to be necessary in this analysis to enable the identification and location of all point discharges and monitoring points necessary for new water use licences application evaluation for water quality allocation in the investigation area. The CMB approach is an indirect approach that provides a viable alternative to other conventional techniques, and it accounts for the benefit of using upstream/downstream river water quality data to estimate the load on the river and identify variations in the water quality characteristics within the river system [50]. An additional advantage of this approach is the substantial reduction in the cost involved in the analysis of a large no of water and effluent samples. The CMB method, as described

by [51–53], was applied in this study. The mass load of a receiving water body is calculated based on Equation (6).

$$QdCd - QuCu = \sum_{i=l}^{n} Li - \sum losses - \sum in\ situ\ generation \tag{6}$$

where $Qd$ and $Qu$ represent downstream and upstream flows, $Cd$ and $Cu$ represent downstream and upstream concentrations in river water, and sum $\sum_{i=l}^{n} Li$ represents the sum of all individual loading into the river if losses and/or generation within the water body are insignificant. In metric units, the concentrations and flows are often expressed in mg/L and $m^3$/s, respectively [51]. Therefore, to determine the resultant impact of a new discharge point in terms of the concentration downstream of a water body, Equation (7) can be applied. In this study, Equation (7) was used to calculate the resultant concentrations of EC, $NO_3$, and $PO_4$ downstream as a result of a new point source discharge. The approach was necessary to assess the impact of a new discharge into the Blesbokspruit River in terms of compliance with the set protection limits for water quality.

$$C_d = \frac{(QuCu + \sum_{i=l}^{n} Li)}{Qd} \tag{7}$$

## 3. Results and Discussion

The water quality index, total maximum daily loads, and waste load allocation for the Blesbokspruit River Catchment are presented and discussed.

### 3.1. Water Quality Index for the Blesbokspruit River Catchment

Table 3 displays the findings of the investigation between 8 and 494 surface water samples for physicochemical characteristics in the research area in comparison to [19,20,37] standards.

**Table 3.** Summary statistics of the physico-chemical parameters determined from the catchment. Mean concentrations (mg/L), pH (standard units), and electrical conductivity (mS/m).

| WQ Variables | Max | Min | Med | Std. Dev | SANS 241: 2015; SAWQG, 1996; WRC, 1998 |
|---|---|---|---|---|---|
| pH | 8.88 | 6.58 | 7.77 | 0.38 | 5–9.7 |
| EC | 808.00 | 12.42 | 86.90 | 47.20 | 170.00 |
| Ca | 136.82 | 23.99 | 35.96 | 37.18 | 150.00 |
| Cl | 169.40 | 4.90 | 66.19 | 25.12 | 300.00 |
| F | 14.00 | 0.03 | 0.27 | 0.79 | 1.50 |
| Mg | 66.42 | 5.75 | 30.06 | 12.75 | 100.00 |
| $NO_3$ | 15.18 | 0.05 | 1.10 | 1.57 | 1.00 |
| Na | 169.08 | 0.05 | 89.40 | 31.46 | 200.00 |
| $SO_4$ | 3873.00 | 5.80 | 219.03 | 390.12 | 250.00 |
| $PO_4$ | 4.20 | 0.03 | 0.43 | 0.52 | 0.025 |

The pH of the surface water in the research area ranged from 6.58 to 8.88 with a median value of 7.77. pH in the research area falls within the limit of the water quality standards. According to [22], the pH is considered one of the main parameters used to establish the alkalinity (pH > 7), acidity (pH < 7), or neutrality (pH = 7) of an environment. Therefore, the findings of this study suggest that surface water in the study area portrays neutral conditions. The EC ranged from 12.42 to 808 mS/m with an average of 86.90 mS/m, and it complies with the limit of the water quality standards. EC is an indicator of salinity [52]; therefore, there is no evidence of salinity problems in the catchment based on the findings of this study.

The concentrations of $Ca^{2+}$ and $Mg^{2+}$ ranged from 23.99 to 136.82 mg/L with an average of 35.96 mg/L for $Ca^{2+}$, and from 5.75 to 66.42 mg/L with an average of 30.06 mg/L for $Mg^{2+}$. The concentrations of the two cations are well within the standards' specified limits.

A median of 89.40 mgL$^{-1}$ was attained for Na$^+$ with a range between 0.05 and 169.08 mg/L, and the concertation falls with the requirement of the standards. The concentrations of F$^-$ and Cl$^-$ range from 0.03 to 14.00 mg/L with an average of 0.27 mg/L for F$^-$ and from 4.90 to 169.40 mg/L with an average of 66.19 mg/L for Cl$^-$. Both anions comply with the standards limits. The water quality results indicate that NO$_3^-$ concentrations averaged 1.10 mg/L with the lowest value 0.05, and the highest value of 15.18 mg/L. The median value for NO$_3^-$ is above the limits specified in the guidelines. The average concentrations of SO$_4^{2-}$ and PO$_4^{2-}$ ranged from 5.80 to 3873 mg/L with an average of 219.03 mg/L for SO$_4^{2-}$. According to [52], SO$_4^{2-}$ is an indicator of acid mine drainage and general mining impacts. Although the average falls below the stipulated limits in the standards, the highest value obtained during the period is alarming. Results for PO$_4^{2-}$ varied from 0.03 to 4.20 mg/L with an average of 0.43 mg/L which is above the required levels prescribed in the water quality standards: PO$_4^{2-}$ is used as an indicator of nutrient enrichment, and potential for eutrophication [53]. This suggests that the land use activities associated with agriculture have a negative impact on the water quality in the study area, thus requiring intervention.

Table 4 provides results of the WQI calculated to establish the overall water quality status on the entire catchment. The index was calculated to be 54.80 indicating, that water in the catchment is good according to WQI ratings (Table 2). According to the South African water quality criteria [20], the overall water quality class of the study area falls in class I, which translates to water suitable for domestic use.

**Table 4.** Water quality index calculated for the research area.

| WQ Variables | Parameter Concentration (Ci) | Standard Limit (Si) | Weight (wi) | Relative Weight (Wi) | Quality Rating Scale (Qi) | Sub-Index (SIi) | WQI |
|---|---|---|---|---|---|---|---|
| pH | 7.77 | 7.35 | 2 | 0.061 | 105.71 | 6.45 | |
| EC | 86.90 | 170.00 | 2 | 0.061 | 51.12 | 3.12 | |
| Ca | 35.96 | 150.00 | 3 | 0.100 | 23.97 | 2.40 | |
| Cl | 66.19 | 300.00 | 4 | 0.133 | 22.06 | 2.94 | |
| F | 0.27 | 1.50 | 4 | 0.133 | 18.00 | 2.39 | 54.80 |
| Mg | 30.06 | 100.00 | 3 | 0.100 | 30.06 | 3.01 | |
| NO$_3$ | 1.10 | 1.00 | 5 | 0.167 | 110.00 | 18.37 | |
| Na | 89.40 | 200.00 | 3 | 0.100 | 44.70 | 4.47 | |
| SO$_4$ | 219.03 | 250.00 | 4 | 0.133 | 87.61 | 11.65 | |

Figures 4–6 show the findings of the historical trend analysis of the physicochemical parameters. The historical trends of Ca$^{2+}$ are depicted in Figure 4a, indicating a steady concentration increase from 2015 to 2020. The concentration of Mg$^{2+}$ (Figure 4b) fluctuated significantly and increased from 2015 to 2022, with the lowest concentration of 23 mg/L and the highest concentration during the period was 48 mg/L. Although the Na$^+$ concentration (Figure 4c) during the same period fluctuated significantly, with the highest levels recorded in 2019 and the lowest recorded in 2021, the trend has not significantly increased. A similar trend is evident for Cl$^-$ (Figure 5a), with the highest concentration recorded in the year 2017. The concentration of NO$_3^-$ (Figure 5b) was the highest in 2015 with significant spikes between 2015 and 2021; however, there has recently been a decline in the concentration. Notably, a significant spike in the concentration of SO$_4^{2-}$ (Figure 5c) was observed between 2017 and 2019. Concentrations of F$^-$ (Figure 6a) and EC (Figure 6c) spiked very strongly in the same year (2018). According to [52], EC and SO$_4^{2-}$ reached undesirable management target limits in the catchment after the Eastern Basin Chemical Acid Mine Drainage treatment plant came into operation. In terms of PO$_4^{2-}$ concentration (Figure 6b), the highest levels were recorded in 2016, 2017, and 2019 and the trend has been slightly declining. The concentration for pH (Figure 6d) was the lowest in 2017 and the highest in 2021, and the trend indicates no significant increase or decline in suggesting that the catchment experiences neutral conditions.

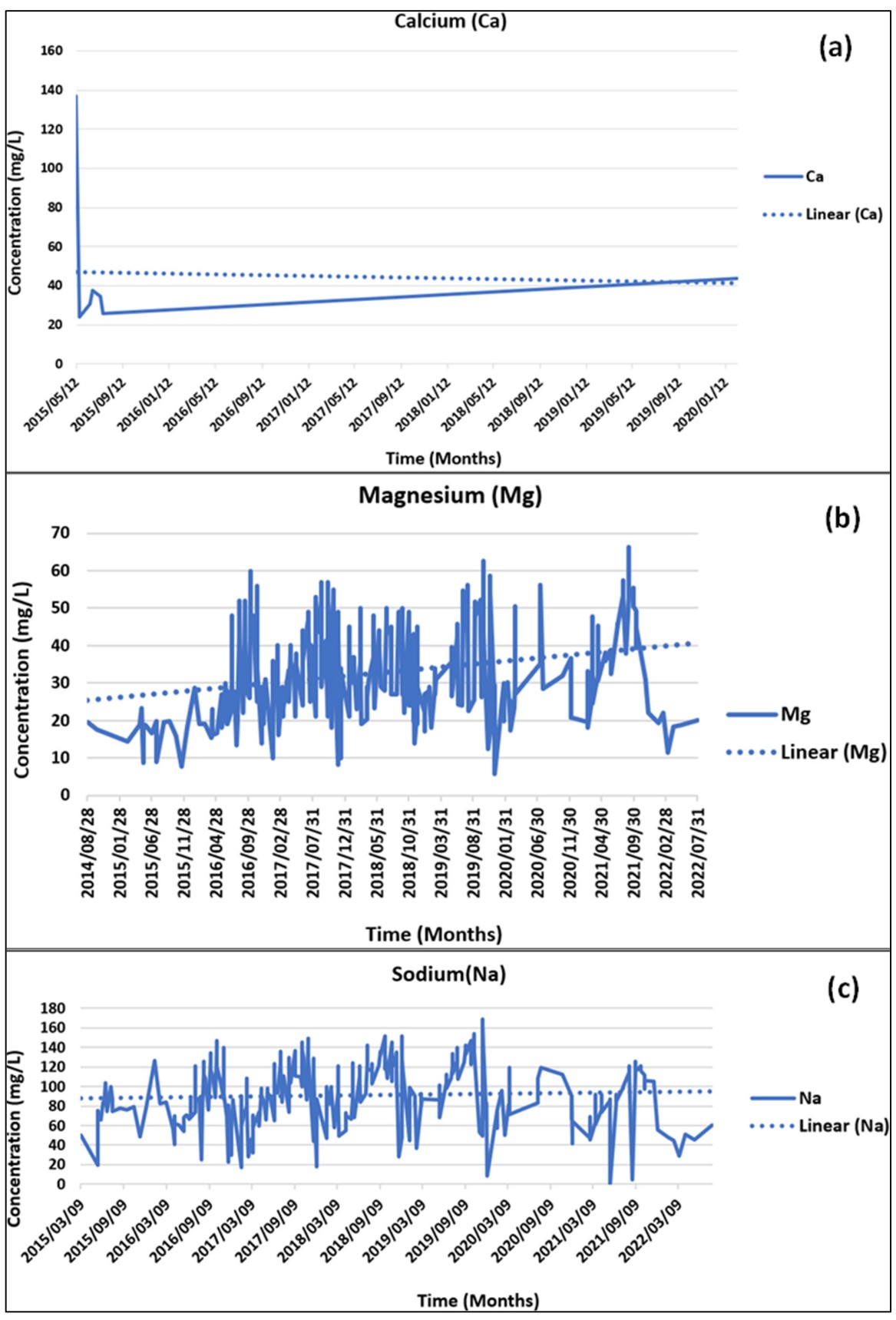

**Figure 4.** Trend analysis of calcium (**a**), magnesium (**b**), and sodium (**c**) from 2015 to 2022 across the study area.

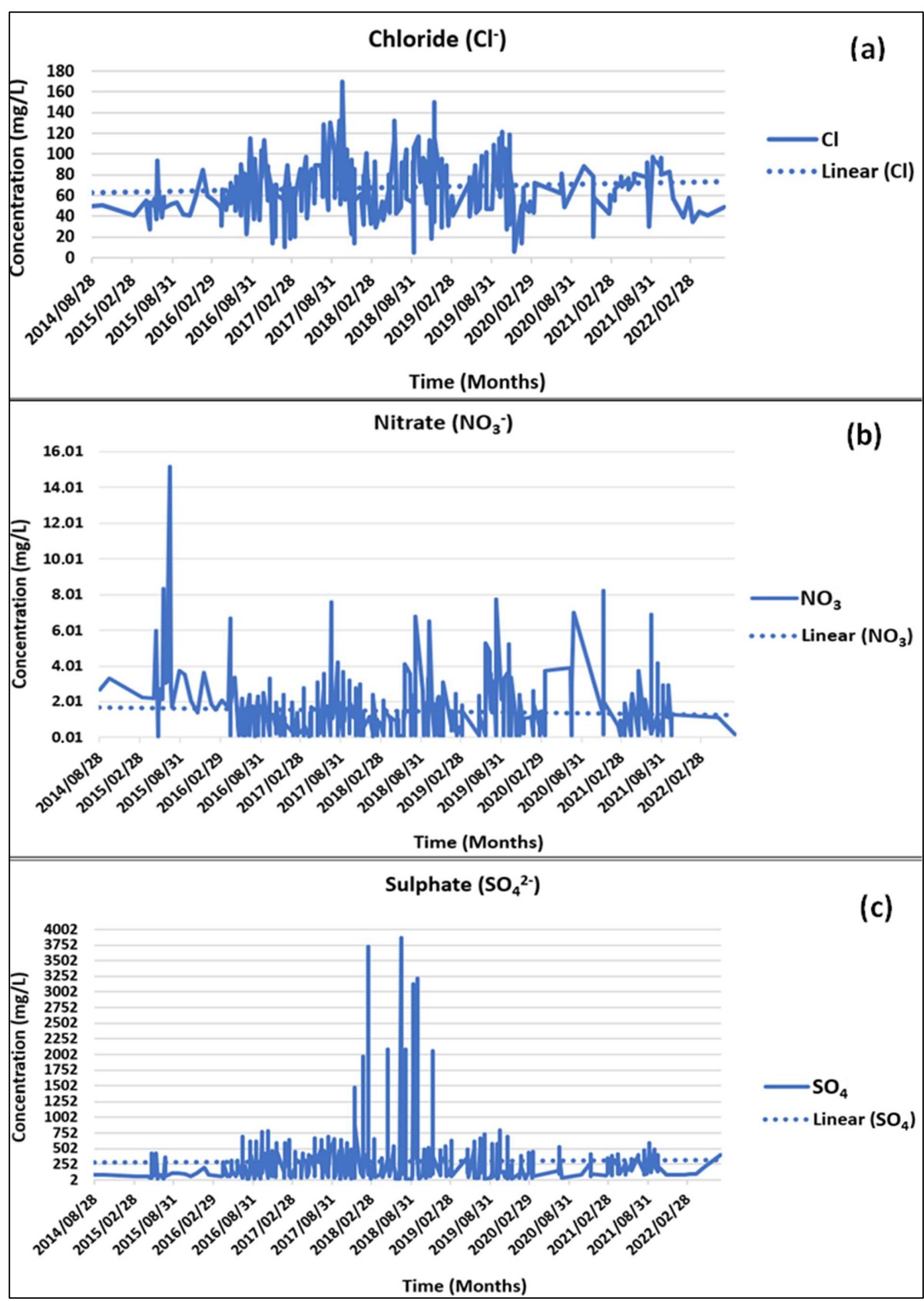

**Figure 5.** Trend analysis of chloride (**a**), nitrate (**b**), and sulphate (**c**) from 2015 to 2022 across the study area.

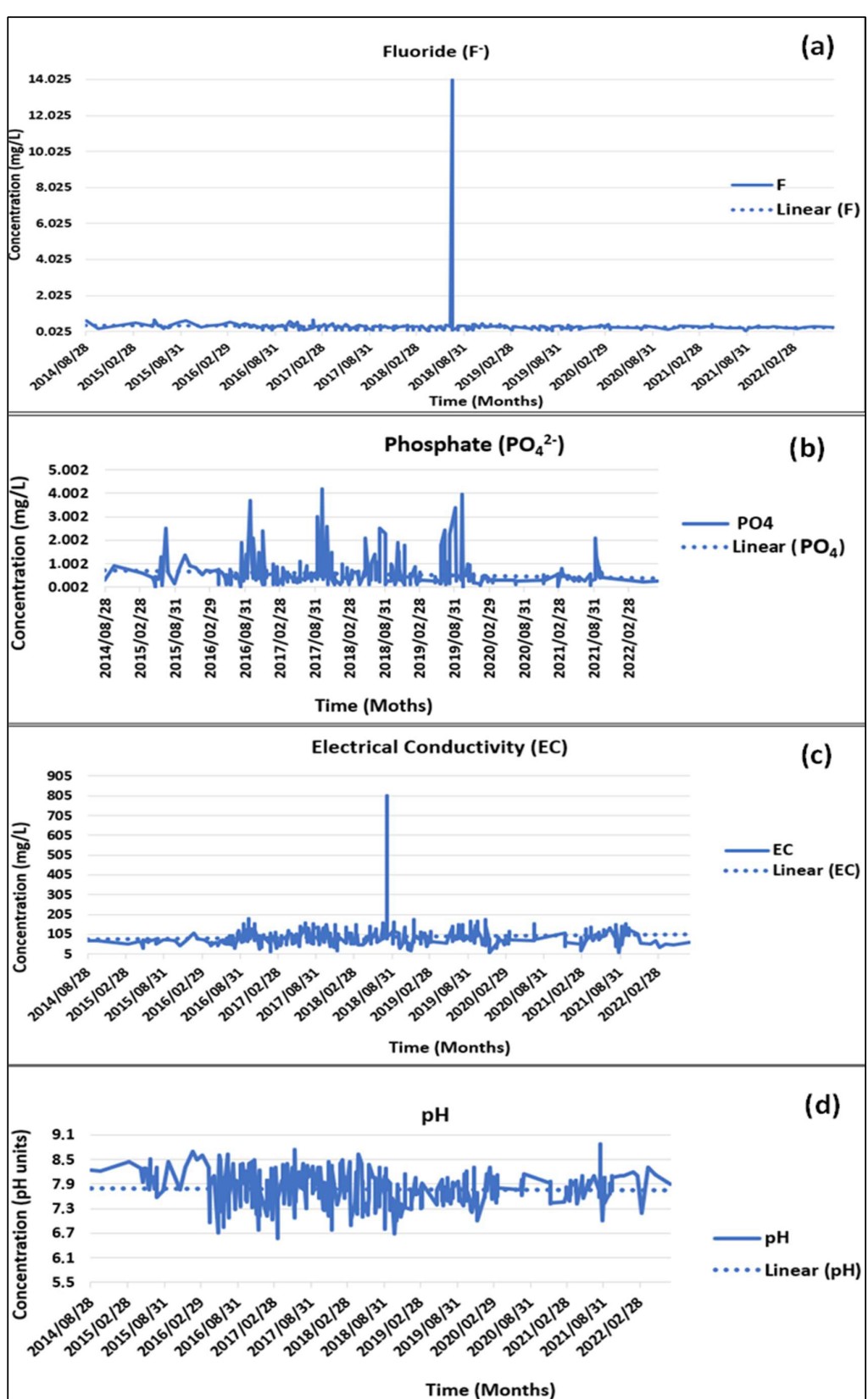

**Figure 6.** Trend analysis of fluoride (**a**), phosphate (**b**), electrical conductivity (**c**), and pH (**d**) from 2015 to 2022 across the study area.

Although results from water quality status assessment using the water quality index indicated that the water quality in the catchment did not adversely deteriorate during the

period from 2015 to 2022. However, the results also gave evidence of $NO_3^-$ and $PO_4^-$ not complying with the standard limits for drinking water, suggesting that the catchment is impacted by agricultural land use activities and wastewater treatment works. $NO_3^-$ and $PO_4^{2-}$ are indicators of sewage and agricultural contamination [53]. The results of the trend analysis for pH, EC, and $SO_4^{2-}$ gave an indication of the mining impact on the catchment which requires intervention. The results concur with earlier findings by [54] which reported that mine dumps in the border of the catchment provide additional source of $SO_4^{2-}$. The result of this study suggests that there is an urgent need for treatment of mine waste before discharge into the stream. Strong compliance monitoring and enforcement with the established water use licences conditions for the user is critical to control pollution from the source. Furthermore, the rehabilitation of the catchment may be considered in the case of exceedance of the protection limits set for the catchment.

### 3.2. Total Maximum Daily Loads for the Blesbokspruit River Catchment

The statistical analysis of water quality data for surface water resource protection in the Blesbokspruit River Catchment is provided in Table 5 for discussion.

**Table 5.** TMDL estimated for the catchment.

| Water Quality Parameter | ΣWLA (mg/s) | ΣLA (mg/s) | MOS (mg/s) | TMDL (mg/s) | Concentration | Gazetted RQOs * |
|---|---|---|---|---|---|---|
| EC | 118,410.00 | 4635.40 | 12,304.54 | 135,349.94 | 77.96 | 180.89 |
| PO$_4$ | 15,653.20 | 9.24 | 1566.24 | 17,228.69 | 9.92 | 3.90 |
| NO$_3$ | 23,796.00 | 129.36 | 2392.54 | 26,317.90 | 15.16 | 10.23 |

Note(s): * All measurements are in mg/L, except for the EC, which is in mS/m [25].

The water quality parameters that we considered for estimation of TMDLs in the study area were EC, $PO_4^{2-}$, and $NO_3^-$ (Table 5). Some of the existing point source load (ΣWLA) was estimated at 118,410.00 mS/m/s, 15,653.20 gm/s, and 23,796.00 mg/s for EC, $PO_4^-$, and $NO_3^-$, respectively. The sum of the load allocation for non-point sources (ΣLA) for EC, $PO4^{2-}$, and $NO_3^-$ was estimated at 4635.40 mS/m/s, 9.24 mg/s, and 129.36 mg/s, respectively. The margin of safety (MOS) was estimated at 12,304.54 mS/m/s for EC, 1566.24 mg/s for $PO_4^{2-}$, and 2392.54 mg/s for $NO_3^-$. The TMDLs for EC, $PO_4^{2-}$, and $NO_3^-$ were estimated at 135,349.94 mS/m/s, 17,228.69 mg/s for $PO_4^{2-}$, and 26,317.90 mg/s for $NO_3^-$. The concentration limits derived from the TMDLs were 77.96 mS/m for EC, 9.92 mg/L for $PO_4^{2-}$, and 15.16 mg/L for $NO_3^-$.

When the results of the study were compared with the findings of the earlier study by [25], undertaken in the same catchment, it gave comparable outcomes of $NO_3^-$ only varying from one another by a magnitude of approximately 5 mg/L. The results of the $PO_4^-$ concentration varied by a magnitude of approximately 6 mg/L. Both these results recorded higher concentration limits from the current study compared to the [25]. However, the concentration of EC derived from the TMDL obtained in the current study was lower that the EC level that had been set by the earlier study, suggesting that the TMDL technique applied in the present study was conservative in terms of EC protection levels for the catchment. The results of this study underline the need for the consideration of existing point and non-point sources when determining protection levels for surface water quality in a catchment which is central to the TMDL approach. The consideration of both point and non-point sources of pollution are critical in the management of land use impact into the resource. Therefore, it is suggested that the total maximum daily loads approach should be employed to establish the protection levels for surface water quality of a resource in a catchment.

### 3.3. Waste Load Allocation for Point Source Discharge

Table 6 provides data that were considered to assess the impacts of a new proposed discharge point to the already established TMDLs. The hypothetical values derived from the catchment for the Upstream Discharge Point 1 were 70.00 mS/m for EC, 3.00 mg/L for

$NO_3^-$, and 0.40 mg/L for $PO_4^{2-}$ with a discharge of 130.00 L/s. The upstream discharge point 2 had hypothetical values of 70.00 mS/m for EC, 15.00 mg/L for $NO_3^-$, 10.00 mg/L for $PO_4^{2-}$, and a discharge of 1560.00 L/s. For upstream discharge point 3, the values were 55.00 mS/m, 3.00 mg/L, 0.60 mg/L, and 2.00 L/s for EC, $NO_3^-$, $PO_4^{2-}$, and the discharge. The TMDLs in terms of concentration for EC, $NO_3^-$, and $PO_4^{2-}$ estimated for the catchment were initially recorded in Table 5 as 70 mS/m, 14.4 mg/L, and 9.24 mg/L, respectively, with a median discharge of 1736.00 L/s recorded at the flow gauging station (B1H032). The new proposed discharge point was intended to discharge 3.00 L/s of waste with concentrations of 85.00 mS/m of EC, 2.30 mg/L of $NO_3^-$, and 6.00 mg/L of $PO_4^{2-}$. The effects of the proposed discharge point to the downstream monitoring point for the catchment resulted in the new concentrations of 68.36 mS/m for EC, 13.71 mg/L for $NO_3^-$, and 9.03 mg/L for $PO_4^{2-}$ estimated using Equation (7). The results indicate that the proposed discharge point resulted in the decrease in the concentrations of EC, $NO_3^-$, $PO_4^{2-}$ in the catchment with new concentration levels being lower than the TMDLs that were set for the catchment as protection limits. This suggests that the proposed discharge point will have beneficial effects on the catchment. Therefore, the proposed discharge point can be allowed to discharge waste with a concentration of 85.00 mS/m of EC, 2.30 mg/L of $NO_3^-$, and 6.00 mg/L of $PO_4^{2-}$ at a discharge rate of 3.00 L/s.

**Table 6.** Existing data from the catchment considered for waste load allocation in the catchment.

| Existing and New Point Discharges | Water Quality | | | Discharge (L/s) |
|---|---|---|---|---|
| | EC (mS/s) | NO3 (mg/L) | PO4 (mg/L) | |
| Upstream discharge point 1 | 70.00 | 3.00 | 0.40 | 130.00 |
| Upstream discharge point 2 | 70.00 | 15.00 | 10.00 | 1560.00 |
| Upstream discharge point 3 | 55.00 | 3.00 | 0.60 | 2.00 |
| *Proposed discharge point* | *85.00 \** | *2.30 \** | *6.00 \** | *3.00 \** |
| [a] TMDL | 70 | 14.4 | 9.24 | 1736.00 |
| *Effects of the proposed discharge point* | *68.36* | *13.71* | *9.03* | |

Note(s): * Hypothetical values for a new discharge point. [a] Values obtained from Table 5.

The evaluation of the impact induced by a new discharge point in terms of the resultant flows in the river indicated that the total discharge was 1.695 $m^3$/s. When Equation (7) was applied to calculate the resultant concentrations of EC, $NO_3^-$, and $PO_4^{2-}$ at a downstream monitoring point because of a new point discharging into the river, the results were as follows. The calculated/resultant concentrations were found to be 70 mg/L for EC, 14.4 mg/L for $NO_3^-$, and 9.24 mg/L $PO_4^{2-}$ (Table 6). The calculated/resultant concentration of 70 mg/L for EC is below the 180.89 mS/m set as the protection limits for the catchment by 61%. Therefore, the new discharge point can be allowed to discharge the proposed 85.00 mS/m concentration of EC into the river. However, the calculated/resultant concentration of 14.4 mg/L for $NO_3^-$ is above the 10.23 mg/L set as the protection limits for the catchment by 41%. In this case, the new discharge point cannot be allowed to discharge the proposed 2.30 mg/L concentration of $NO_3^-$ as this will results into the protection levels set for the catchment being exceeded. In terms of $PO_4^{2-}$, the calculated/resultant concentration of 9.24 mg/L far exceeds (137%) the protection limits of 3.9 mg/L set for the catchment. Therefore, the new discharge point cannot be allowed to discharge the proposed 6.00 mg/L of $PO_4^{2-}$; instead, the waste will have to be treated to lower concentration levels before discharge, otherwise the protection levels set for the catchment would be compromised.

## 4. Conclusions

The operationalization of the water resource protection initiatives within the Blesbokspruit River Catchment was assessed. The study discovered that the overall water quality status for the physico-chemical parameters ($Na^+$, $Ca^{2+}$, $Mg^{2+}$, $Cl^-$, $F^-$, pH, EC, $SO_4^{2-}$, $PO_4^-$) is within the limits of the water quality standards, except for $NO_3$- and $PO_{4-}$. The concentration levels of parameters including EC, $SO_4^{2-}$, $NO_3^-$, and $PO_4^{2-}$ vary significantly as a result of the mining activities, and waste discharge from wastewater treatment works. The application of the TMDL and CMB approaches facilitates water

resource protection practices at the catchment level. Continuous water resource monitoring for surface water quality is critical for compliance monitoring against established protection levels in a catchment. The findings of this study provide critical evidence on the feasibility of resource-directed measures' implementation at the catchment level for water resources' protection in South Africa. To improve water resource protection practices at the catchment level, this study recommends the application of TMDL and CMB techniques, and that active adaptive management actions should form part of any water resource management plans in a catchment. The study also recommends further research into the application of adaptive management tools such as treatability index techniques for improved water resource protection practices.

**Author Contributions:** K.M. was responsible for data collection, data analysis, and drafting the manuscript. S.N. and T.K. were responsible for the review and editing of the manuscript. S.N. is the academic co-supervisor of the corresponding author and T.K. is the academic supervisor of the corresponding author. All authors have read and agreed to the published version of the manuscript.

**Funding:** This research was funded by the Department of Water and Sanitation (South Africa).

**Data Availability Statement:** Not applicable.

**Acknowledgments:** This paper is part of a Ph.D. project by the corresponding author. The author gratefully acknowledges the Department of Water and Sanitation (DWS), Water Ecosystem Management and Resource Quality Information Services (RQIS) for availing the historical data.

**Conflicts of Interest:** The authors declare no conflict of interest.

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
