# Peer review of "Assessing Feasibility of Water Resource Protection Practice at Catchment Level: A Case of the Blesbokspruit River Catchment, South Africa"

_water, doi:10.3390/w15132394_

Round 1

Reviewer 1 Report

Comments to Authors:

In this study, the authors performed feasibility assessment of water resource protection measures operationalization: Implications for surface water quality protection and allocation in the Blesbokspruit River Catchment of the South Africa. Overall, the study is methodologically sound with promising results. However, prior to further consideration, some comments to be addressed:

(i) Title is slightly confusing. I would suggest shortening the title.

(ii) Introduction: Huge modifications are necessary for the introduction section. The research gaps and significance of the study are not shown. Moreover, the authors should provide more precise research objectives in this study.

(iii) Fig 1. Can we see the land use? If yes [please consider include it.

(iv).  There are so many groundwater WQI available, but why this WQI is adopted? Explain with reasons. What about South Africa own GWQI?

(v). Table 1. The table only provide measurements from Class 0 to Class III but the WQI ratings is having five classifications. I would remake the classification to 4 groups instead of 1. Please refer to this article on how the authors made the reclassification. (i) Comparison among different ASEAN water quality indices for the assessment of the spatial variation of surface water quality in the Selangor river basin, Malaysia. (ii) Application of artificial intelligence methods for monsoonal river classification in Selangor river basin, Malaysia

(vi) Provide a flow chart in the methodology. Include some introduction about the system. Move it from the results and discussion to methodology. The methodology is not well understood in this study.

(vii) Table 3 is good, but will be better if you can replace it into a bar charts for better illustration.

(viii) How the weight is assigned in the table 4?

(ix) Discussion is lacking in this study. So what are the proposed mitigations for better protection? What should the government do for proper water quality management? Effects from changing land use or anthropogenic activities? The difference between upstream or downstream is not provided.

(x).  Conclusion section seems to be a repetition of the results section. Huge modifications are required. Please provide insights into this study and what can be further done in the future.

(xi).  Implications for future research may also be included in the conclusion at the end.

Proofreading should be performed.

Author Response

response for reviewer no 1 has been uploaded

Reviewer 2 Report

Authors did a great job to execute this work especially from this data scarce region. This work can be accepted for publication once they consider following suggestions:

1. For table 2, definition categorization is not so clear. Ex. What do you mean by ideal? or Acceptable water for what? It all depends on who is the end consumer? Please revise

2. Although you have highlighted the issues/challenges, there are no policy intervention for better water resource management  

It is fine for me

Author Response

Response to reviewer no 2 has been uploaded

Reviewer 3 Report

The paper can be accepted with the following revisions.

1.      Figure 1 is unclear.

2.      More details about figure 6 should be provided.

3.      The information’s reported in Table 6 are unclear.

4.      The authors should provide more details about equation 5 and how it is reported in the literature.

5.      Quality of figure 3 should be improved.

Author Response

response to reviewer no 3 has been uploaded

Round 2

Reviewer 1 Report

Comments to the authors:

The authors have substantially addressed comments that I raised during the first review, however, since you adopted the idealogy from (i) Comparison among different ASEAN water quality indices for the assessment of the spatial variation of surface water quality in the Selangor river basin, Malaysia. (ii) Application of artificial intelligence methods for monsoonal river classification in Selangor river basin, Malaysia, please cite them accoringly.

Good job in revision!

minor english editings will be required.

Author Response

response to reviewer uploaded
